# THE SEMANTIC HUB HYPOTHESIS: LANGUAGE MODELS SHARE SEMANTIC REPRESENTATIONS ACROSS LANGUAGES AND MODALITIES

**Zhaofeng Wu**[@] **Xinyan Velocity Yu**[II] **Dani Yogatama**[II] **Jiasen Lu**[=] **Yoon Kim**[@]
[@]MIT  [II]University of Southern California  [=]Allen Institute for AI
zfw@csail.mit.edu

## ABSTRACT

Modern language models can process inputs across diverse languages and modalities. We hypothesize that models acquire this capability through learning a *shared representation space* across heterogeneous data types (e.g., different languages and modalities), which places semantically similar inputs near one another, even if they are from different modalities/languages. We term this the *semantic hub hypothesis*, following the hub-and-spoke model from neuroscience (Patterson et al., 2007) which posits that semantic knowledge in the human brain is organized through a transmodal semantic "hub" which integrates information from various modality-specific "spokes" regions. We first show that model representations for semantically equivalent inputs in different languages are similar in the intermediate layers, and that this space can be interpreted using the model's dominant pretraining language via the logit lens. This tendency extends to other data types, including arithmetic expressions, code, and visual/audio inputs. Interventions in the shared representation space in one data type also predictably affect model outputs in other data types, suggesting that this shared representations space is not simply a vestigial byproduct of large-scale training on broad data, but something that is actively utilized by the model during input processing.

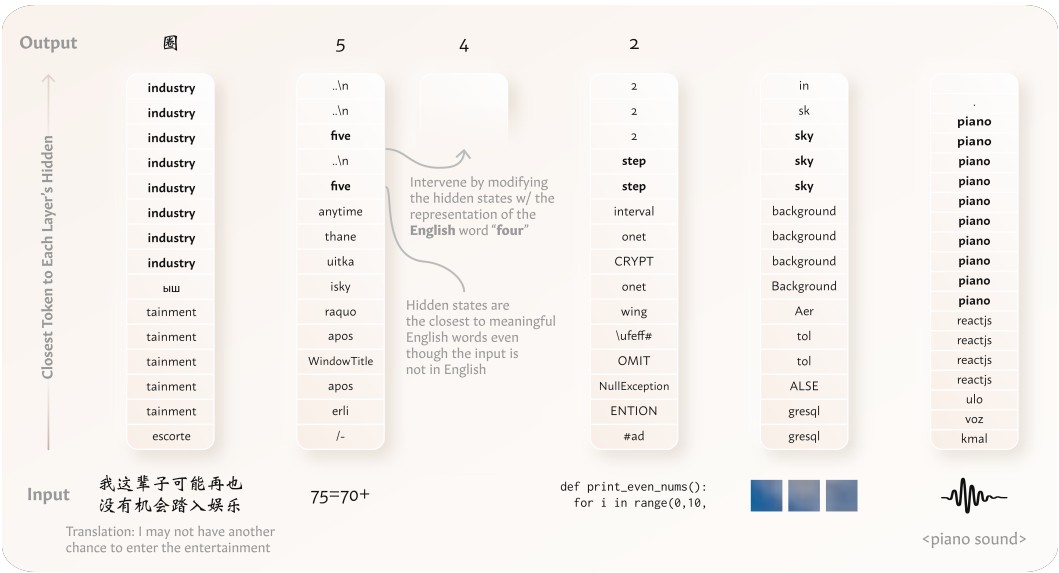

**Figure 1:** Examples of the semantic hub effect across input data types. For every other layer, we show the closest output token to the hidden state based on the logit lens. Llama-3's hidden states are often closest to English tokens when processing Chinese texts, arithmetic expressions (in §C in the appendix), and code, in a semantically corresponding way. LLaVA, a vision-language model, and SALMONN, an audio-language model, have similar behavior when processing images/audio. As shown for the arithmetic expression example, models can be intervened cross-lingually or cross-modally, such as using English even though the input is non-English, and be steered towards corresponding effects. Boldface is only for emphasis.

# 1 INTRODUCTION

Modern language and multimodal models (LMs)[1] can process heterogeneous data types: text in different languages, non-linguistic inputs such as code and math expressions, and even other modalities such as images and sound. How do LMs process these distinct data types with a single set of parameters? One strategy might be to learn specialized subspaces for each data type that are only employed when processing it. Often, however, surface-distinct data types share underlying semantic concepts. This is most obvious for sentences in different languages with the same meaning; but such shared concepts are present across other data types, e.g., between an image and its caption, or a piece of code and its natural language description. The human brain, for example, is believed to have a transmodal "semantic hub" (Patterson et al., 2007; Ralph et al., 2017) that integrates semantic information from modality-specific "spokes" (e.g., visual/auditory cortices). A model, leveraging structural commonalities across data types, could similarly project their surface forms into a *shared* representation space, perform computations in it, and then project back out into surface forms.

To what extent is this idealized strategy adopted by actual models? Wendler et al. (2024) find that on simple synthetic tasks, Llama-2 (Touvron et al., 2023b) maps various input languages into a shared "English space", hinting that it leverages this shared representation scheme to an extent. We show that this is in fact a much more general phenomenon: **when a model processes inputs from multiple data types, there is a shared representation space, and this space can be interpreted to an extent in the LM's inherently dominant language (usually English)** via the logit lens (nostalgebraist, 2020). Concretely, we say that this space is *scaffolded* by tokens in the LM's dominant language.

We first show that LMs represent semantically similar inputs from distinct data types (across languages, or between natural language and arithmetic expressions, code, formal semantic structures, and multimodal inputs) to be close to one another in intermediate LM layers. We further show that we can interpret these hidden representations to an extent using the LM's dominant data type—e.g., when processing a Chinese input, an English-dominant LM "thinks" in English before projecting back out to a Chinese space. Finally, we perform intervention experiments showing that intervening in the shared representation space using the LM's dominant data type, predictably affects model output when processing other data types; that is, the shared representation space (and the processing of these representations through subsequent layers) is not a vestigial byproduct of the model's being trained on (say) English-dominant text, but causally impacts model behavior.

Our work is complementary and distinct from prior work which finds structural similarities between the representation spaces of models trained (usually independently) on different data types, such as those showing that text representations from text-only LMs can be aligned, via a transformation, to vision/audio representations of modality-specific models (Ilharco et al., 2021; Merullo et al., 2022; Li et al., 2023; Ngo & Kim, 2024; Huh et al., 2024; *i.a.*), the literature on cross-lingual word embedding alignment (Mikolov et al., 2013; Artetxe et al., 2017; Conneau et al., 2018; Schuster et al., 2019; *i.a.*), and work on cross-task transfer (Moschella et al., 2023; Wu et al., 2024; *i.a.*). We instead show that an LM trained on multiple data types represents and processes them in a shared space *without* requiring explicit alignment transformation. We hope our findings inspire ways to more easily interpret the mechanisms of current models and future work aimed at better controlling models.

# 2 THE SEMANTIC HUB HYPOTHESIS

Let $\mathcal{X}_z$ be the domain of a data type $z \in \mathcal{Z}$ where $\mathcal{Z}$ denotes all model-supported data types. E.g., for languages, $\mathcal{X}_{\text{Chinese}}$ could be all Chinese tokens, while for images $\mathcal{X}_{\text{Image}} = [0, 255]^{w \times h \times 3}$ could be the RGB values for an $w \times h$-sized patch. Consider a function $M_z : \mathcal{X}_z^* \to \mathcal{S}_z$ mapping an input sequence into a semantic representation space $\mathcal{S}_z$ (the "hub"), and a verbalization function $V_z : \mathcal{S}_z \to \mathcal{X}_z^*$. Given an input prefix $w_{1:t}^z \in \mathcal{X}_z^*$ of length $t$ where $w_i^z \in \mathcal{X}_z$, a sensible (implementation-agnostic) way to continue the sequence is to first encode it into a modality-agnostic representation, $m^{in} = M_z(w_{1:t})$, reason and formulate a representation of futures $m^{out} \in \mathcal{S}_z$, and finally verbalizing via $V_z(m^{out})$.

An LM parameterizes a similar process: it uses $M_{\text{LM}}$ to map various input data types into a representation space $\mathcal{S}_{\text{LM}} \subseteq \mathbb{R}^d$ (early layers), performs computations in the space (middle layers), and verbalizes the output via $V_{\text{LM}}$ (end layers and the LM head). However, it is unknown how different

---

[1]Hereafter, we use the term "language model" loosely and also consider multimodal language models that process additional data modalities, since such models are commonly trained on top of a text LM backbone.

data types are structured in the representation space. E.g., one possibility is that the LM partitions $\mathbb{R}^d$ into disjoint subspaces for each data type and processes them separately. We instead hypothesize that LMs learn to represent and process different data types in a *shared* representation space that functions as a modality-agnostic "semantic hub." That is, semantically similar inputs $w_{1:t}^{z_1}$ and $w_{1:t'}^{z_2}$ from distinct data types—e.g., texts in different languages that are mutual translations—are similarly mapped in $\mathcal{S}_{\text{LM}}$; informally, $M_{\text{LM}}(w_{1:t}^{z_1}) \approx M_{\text{LM}}(w_{1:t'}^{z_2})$. However, absolute similarity measures (i.e., $\text{sim}(M_{\text{LM}}(w_{1:t}^{z_1}), M_{\text{LM}}(w_{1:t'}^{z_2}))$) are generally difficult and unintuitive to interpret in high dimensional spaces.[2] We thus focus on relative similarity measures, taking a semantically unrelated sequence $u_{1:t'}^{z_2}$, and evaluating whether $w_{1:t}^{z_1}$ is closer to $w_{1:t'}^{z_2}$ than $u_{1:t''}^{z_2}$. We formalize this hypothesis as:

$$\text{sim}\left(M_{\text{LM}}(w_{1:t}^{z_1}), M_{\text{LM}}(w_{1:t'}^{z_2})\right) > \text{sim}\left(M_{\text{LM}}(w_{1:t}^{z_1}), M_{\text{LM}}(u_{1:t''}^{z_2})\right). \tag{1}$$

Moreover, when the LM has a *dominant data type* $z^\star$ in training (e.g., English for Llama-2), we hypothesize that this shared representation space is "anchored" by $z^\star$, in the sense that Eq. 1 holds strongly enough that we can probe out to $z^\star$ from $M_{\text{LM}}(w_{1:t}^z)$. We further expect this to hold for model representations of the *future*, which autoregressive LMs are trained to model. I.e., the representation of a prefix should better align with a verbalization of the future in $z^\star$ than a non-dominant $z^\circ$ (though we now need a different kind of model representation, denoted with $\text{repr}_{\text{LM}}$):

$$\text{sim}\left(M_{\text{LM}}(w_{1:t}^z), \text{repr}_{\text{LM}}(w_{>t}^{z^\star})\right) > \text{sim}\left(M_{\text{LM}}(w_{1:t}^z), \text{repr}_{\text{LM}}(w_{>t}^{z^\circ})\right). \tag{2}$$

We hypothesize that this holds even when $z = z^\circ$. E.g., with an English-dominant LM, its encoding of the Chinese prefix $w_{1:t}^\circ =$"这篇论文太难" (trans. "This paper is so hard to") should be closer to the representation of the English word $w_{t+1}^{z^\star} =$"write" than its Chinese translation $w_{t+1}^{z^\circ} =$"写".

## 2.1 METHOD: TESTING THE SEMANTIC HUB HYPOTHESIS

We test the semantic hub hypothesis by considering pairs of distinct data types, the dominant one $z^\star$ and a non-dominant one $z^\circ$, different for each experiment. Whenever semantically related inputs are available (e.g., an image and its caption), we directly test Eq. 1 by using $h_t^\ell$, the LM's hidden state at position $t$ and layer $\ell$, as $M_{\text{LM}}(w_{1:t}^z)$, and further using cosine similarity for the similarity function.

We operationalize Eq. 2 via the *logit lens* (nostalgebraist, 2020), a simple training-free approach for interpreting the hidden states of a model. Transformer-based LMs produce the next-token distribution using $\text{softmax}\left(Oh_t^L\right)$ (omitting the bias term) where $O$ is the output token embeddings (or "unembeddings") and $h_t^L$ is the final layer hidden state. Logit lens applies the same operation to the intermediate layers to obtain $p^{\text{logitlens}}(\cdot \mid h_t^\ell) := \text{softmax}\left(Oh_t^\ell\right)$. Logit lens has been found to produce meaningful distributions that shed light on an LM's internal representations and computations.

Under the logit lens, $\text{repr}_{\text{LM}}$ in Eq. 2 considers the output embedding of single tokens, $\tau^{z^\star} \in \mathcal{X}_{z^\star}$ and $\tau^{z^\circ} \in \mathcal{X}_{z^\circ}$. We use the first token of $w_{>t}^{z^\star}$ and $w_{>t}^{z^\circ}$. Using the dot product for $\text{sim}(\cdot)$, Eq. 2 is equivalent to comparing the logit lens probabilities,

$$p^{\text{logitlens}}\left(\tau^{z^\star} \mid h_t^\ell\right) > p^{\text{logitlens}}\left(\tau^{z^\circ} \mid h_t^\ell\right), \tag{3}$$

i.e., testing whether the probability of the continuation in the dominant language is more likely than the continuation in the original input data type. Since the logit lens is tailored for probing out a single token, we usually consider short-enough verbalizations such that a single BPE token can reliability identify it. This often means that the two verbalizations are two single words that are semantic equivalents. Nevertheless, we also consider longer future verbalizations when its first token unambiguously suggests one interpretation in that context, which allows more flexibility.

While the above test is simple, $\tau^{z^\circ}$ is unavailable in many multimodal models without vocabulary tokens for $z^\circ$. We thus only focus on testing Eq. 1, though logit lens enables an additional test:

$$p^{\text{logitlens}}\left(\tau^{z^\star} \mid h_t^\ell\right) > p^{\text{logitlens}}\left(\upsilon^{z^\star} \mid h_t^\ell\right), \tag{4}$$

where $\upsilon$ is an unrelated token. Also, using the *next* token as $\tau^{z^\star}$ is unsuitable for multimodal models never been trained to output at non-dominant data type positions $t$, but the non-language data types are

---

[2]See for example Beyer et al. (1999). Most prior work in the probing literature also implicitly uses relative similarity measures since the similarity scores are normalized over a finite label set.

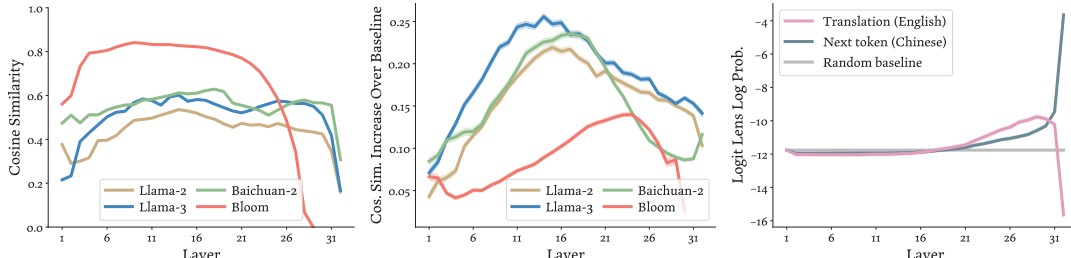

**(a)** The cosine similarity of intermediate representations of English and Chinese parallel texts.

**(b)** Same as (a), but subtracted by a baseline over non-parallel texts.

**(c)** Llama-3 logit lens log prob. of parallel English vs. Chinese tokens when processing Chinese text.

**Figure 2:** Results for the multilingual experiments. The 95% CI is plotted in all. **Parallel texts have similar representations. Hidden states for Chinese texts are close to the unembedding of English tokens.**

only encoded as prefixes. For such models, we hypothesize that they still fully represent the *current* input in a shared space. We thus take $\tau^{z^\star}$ to be the last token of $w_{1:t'}^{z^\star}$ (which is the interpretation of $w_{1:t}^{z^\circ}$ under $z^\star$; e.g., if $w_{1:t}^{z^\circ}$ is an image, $w_{1:t'}^{z^\star}$ can be the objects in the image described in language).

## 3 EVIDENCE OF A SEMANTIC HUB

We apply our tests across diverse data types and find evidence of a shared representation space in all cases. We show additional experiments on arithmetic data in §C with similar trends.

### 3.1 MULTILINGUAL

Wendler et al. (2024) find that when processing specific in-context learning (ICL) templates for highly synthetic lexical-level tasks (word repetition, word translation, etc.) in non-English languages, the intermediate hidden states of Llama-2 are closer to the unembeddings of English tokens than the output language. This is consistent with our hypothesis, albeit constrained to a simple synthetic task and one LM. We show that this shared representation space is a general property of LMs when they face naturally occurring text.

**Experiment 1: Representation similarity of mutual translations.** Translation datasets enable a direct test for Eq. 1, with semantically equivalent cross-lingual sentences as $w_{1:t}^{z_1}$ and $w_{1:t'}^{z_2}$ and a randomly chosen non-matching sentence as $u_{1:t''}^{z_2}$. We use the professionally-translated English-Chinese parallel sentences from Chen et al. (2016) ($N = 5260$). For each sentence pair, we use a template to transform each sentence and compute the representation cosine similarity for each layer, using the last token position as the sentence representation following Wu et al. (2023), which preserves sentence information (Morris et al., 2023). We consider two English-dominant LMs, Llama-2 and Llama-3, one Chinese-dominant LM, Baichuan-2 (Yang et al., 2023), and one multilingual LM, BLOOM (BigScience, 2023), specifically the 7B/8B variants. §A.1 contains more details.

In Figure 2a, the raw cosine similarity is high, up to >80% (left), and it is also significantly higher than the non-matching pairs' similarities (right), but only in the middle layers. These trends support the hypothesis that the middle layers act as the semantic hub of the LM. Notably, this trend also exists for BLOOM which does not have a dominant pretraining language.

**Experiment 2: Probing out continuations in the dominant language.** We next test Eq. 3: whether continuations in the dominant language have *higher* probability than those in the input language. We use 1,000 Chinese and English sentences from Wikipedia (Wikimedia-Foundation, 2023). For the English-dominant LLama-3, we use a Chinese prefix $w_{1:t}^{z^\circ}$ as input and take $\tau^{z^\circ}$ to be the next Chinese token (i.e., $w_{t+1}^{z^\circ}$) and $\tau^{z^\star}$ to be the (first token of the) English translation of $w_{t+1}^{z^\circ}$. §A.1 has further details. Figure 2c plots the logit lens probability for the two tokens as well as the uniform distribution probability. In early layers, we cannot read out either token better than random chance. After layer 17, the model representations are substantially closer to the English token than the Chinese token until layer 31, showing that the model hidden space is indeed better scaffolded by English than Chinese.

Next, we extend this analysis to consider global language-level trends across languages. We first compute $p(w \mid z)$, the token distribution under a language $z$, by running the LM tokenizer on the language-specific split of the mC4 dataset (Xue et al., 2021). We then use Bayes' rule to estimate

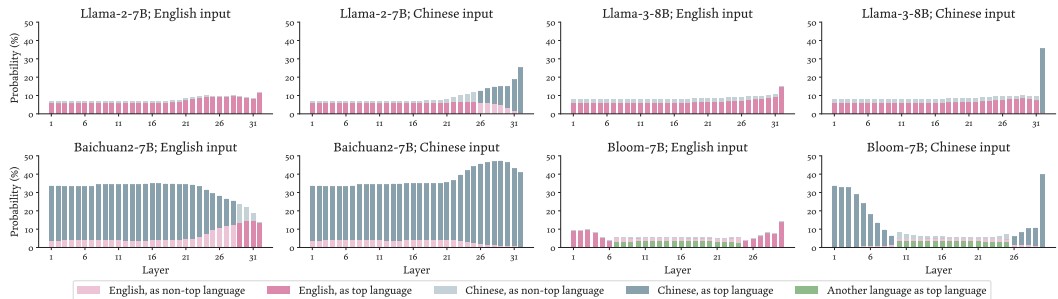

**Figure 3:** Language prob. of English and Chinese (and the top language when it is neither, which only happens for Bloom). **Regardless of the input language, the dominant LM language is more salient in early-middle layers, and the input language is more salient in the final layers. Bloom has no clear intermediate language.**

$p(z \mid w) \propto p(w \mid z)p(z)$ with a uniform prior $p(z)$.[3] We finally compute the probability of $h_t^\ell$ belonging to each language as $p(z \mid h_t^\ell) \propto \sum_{w \in \mathcal{V}} p(z \mid w)p^{\text{logitlens}}(w \mid h_t^\ell)$. If our hypothesis that the universal representation space is better scaffolded by the dominant language is true, we expect the dominant language $z^\star$ to have the highest probability across input languages in the middle layers.

Figure 3 shows the layer-wise probability of English and Chinese on 10,000 English/Chinese Wikipedia sentences. When English-dominant models process Chinese text, Wendler et al.'s finding generalizes, where English dominates in intermediate layers and Chinese only dominates in the final layers. On the Chinese-dominant LM, this trend flips: when processing English text, its intermediate layers are closer to Chinese tokens and the final layers are closer to English tokens. For BLOOM, a multilingual model with a relatively balanced training language mixture, we do not see a clear dominating language in intermediate layers; when we manually inspect the closest tokens, in most cases they are symbols with no clear semantics (though this does not mean it lacks a unified representation space: see Exp. 1). 70B model trends in §A.1 are highly similar to the 7B/8B ones.

## 3.2 CODE

Many recent LMs are trained on code corpora (Touvron et al., 2023a;b; Llama-3-Team, 2024; Gemini-Team, 2024). We find that they similarly process code by projecting it into a unified representation space shared with regular language tokens. Figure 4 shows examples, where LMs in the intermediate layers tend to verbalize the future in free-form English, unconstrained by program syntax. E.g., in the first program, given the Python prefix "... for idx, elem in enumerate(numbers): for idx2, elem2 in enumerate(numbers)", instead of the groundtruth continuation in Python "): if idx != idx2: ...", the most salient intermediate token is "except", likely attempting to predict in English "(for each element in numbers) except if it is equal to idx". Similarly, in

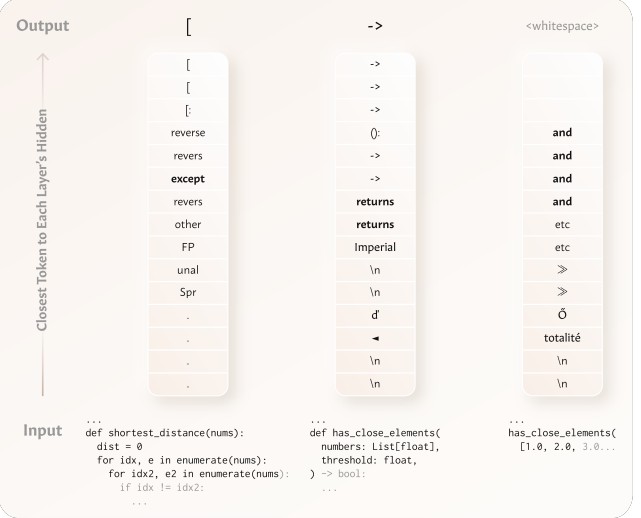

**Figure 4:** Logit lens analysis on Llama-2 processing Python programs. For every other layer, we show the closest token (sometimes whitespace) to the hidden states *before* the grayed-out texts. **The model tends to verbalize the future prediction in English that corresponds to the code continuations (in gray).**

a list expression "[1.0, 2.0," instead of continuing in Python " 3.0", it predicts "and", which is a natural way to continue in English. In these cases, it is difficult to obtain semantically equivalent English-Python pairs, so we only test Eq. 3 across targeted cases in Python below.

**Experiment 1: Simple Python list literals.** We systematically test the list case, where $h_t^l$ is the hidden state after processing ",", $\tau^{z^\star}$ ="and", and $\tau^{z^\circ}$ is the actual next token. Figure 5a shows

---

[3]This prior obviously does not reflect the training language distribution, but in fact makes our trends even more salient, since using a real (or estimated) $p(z)$ would make $p(z \mid w)$ even larger for the dominant language.

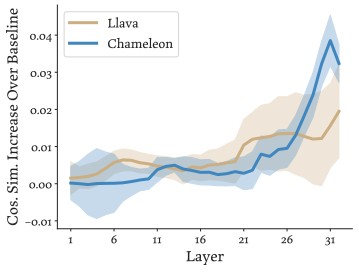

**(a)** Llama-2 logit lens log probabilities (and the 95% CI) at commas in Python list literals, of the English "and" token (and baseline tokens) vs. the actual next token in the program, from MBPP.

**(b)** The distance between Llama-2 hidden states when predicting a function argument, to the unembedding of the argument's name (its semantic role) vs. the actual argument expression, in MBPP.

**Figure 5:** Results for the code experiments. **Code expressions are close to semantically meaningful free-form English words in early-middle layers, such as "and" in list literals and the argument's semantic role in function calls; in the final layers, the representation converges to the context-constrained Python token.**

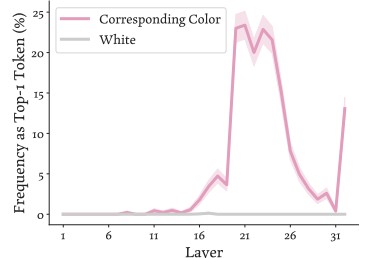
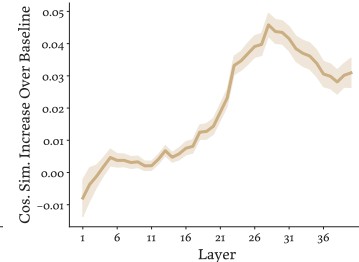

**(a)** The cosine similarity difference between intermediate representations of matching images and captions, over non-matching ones.

**(b)** The frequency of the closest token to LLaVA's hidden states describing the image color, against a baseline using "white".

**(c)** The cosine similarity difference between intermediate SALMONN representations of matching audios and labels, over non-matching ones.

**Figure 6:** Results for the multimodal experiments. The 95% CI is plotted in all. **Model representation of (a) visual and (c) audio inputs and their textual labels are similar. Furthermore, in (b), model representations of individual color patch tokens are close to English words for those colors.**

that this trend holds systematically on all such commas in the MBPP dataset (Austin et al., 2021) ($N = 6923$, including unit tests): as expected, in the final layers, the representation is closer to the ground truth next token's unembedding, and closer to "and" in the middle layers. We also show the probability with two other tokens, "or" and "not", as baselines, both of which are lower than "and".

**Experiment 2: Python function call arguments.** Function arguments have names in the definition, such as "range(start, end, step)"; but when invoked, they are filled with actual context-appropriate expressions. We call the argument names "semantic roles", and the context-specific expressions the "surface forms", inspired by thematic relations in linguistics (Fillmore, 1968). In the second example in Figure 1, we show that LMs predict the arguments by first "thinking" about their semantic role ($\tau^{z^\star}$) and then instantiating with surface-constrained expressions ($\tau^{z^\circ}$). We extract all function calls and arguments from MBPP with simple filtering, resulting in 540 arguments (see §A.2 for details). For each argument, we use the logit lens to inspect the hidden states at the preceding token ("(" or ","). For each argument, Figure 5b visualizes if the semantic role or the surface form is closer to each layer's hidden state of Llama-2. The semantic role ($\tau^{z^\star}$) dominates for the early to middle layers, even though the role token usually does not appear in the context at all, and only in the final layers do the representations converge towards the surface form argument ($\tau^{z^\circ}$).

## 3.3 VISUAL INPUT

Past work has investigated the representation of *separately trained* vision and text models, often finding that their representation spaces are similarly structured and alignable (Merullo et al., 2022; Li et al., 2023; Huh et al., 2024; *i.a.*). We show that when trained together, vision-language models learn to project both modalities into a joint representation space. Current vision-language models typically represent images by segmenting them into patches, embedding them into "image tokens",

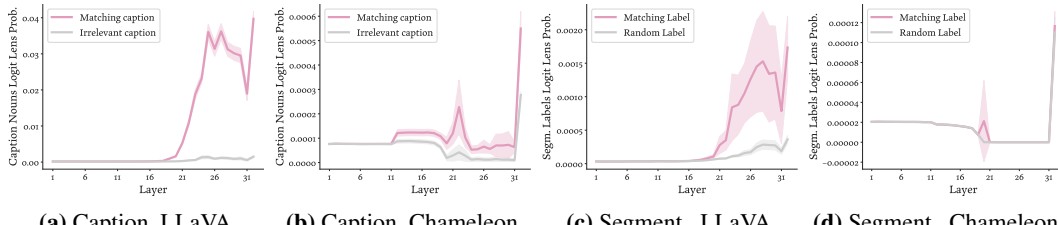

**(a)** Caption, LLaVA    **(b)** Caption, Chameleon    **(c)** Segment., LLaVA    **(d)** Segment., Chameleon

**Figure 7:** When processing an image patch, model logit lens probabilities of either the nouns in the corresponding caption or the patch segmentation label, as well as a baseline for each with no correspondence between the patch and the label. **The image representations better match the semantically corresponding English words.**

and then feeding them into the transformer model along with other text tokens (Lu et al., 2023; 2024; Liu et al., 2023; *i.a.*). We hypothesize that the intermediate representations of the image patches are close to the corresponding language tokens that describe the scene. Experimental details are in §A.3.

**Experiment 1: Representation similarity between an image and its caption.** Though not constituting exact semantic equivalence, an image paired with its caption provides one possible test for Eq. 1. We take 1000 images and corresponding captions in the MSCOCO dataset (Lin et al., 2014) and measure their hidden states cosine similarity in LLaVA-7B (Liu et al., 2023) and Chameleon-7B (Chameleon-Team, 2024). As in Eq. 1, we subtract the average cosine similarity between non-matching image-caption pairs as a baseline, separately for each layer. Figure 6a shows that semantically matching inputs, even though in different modalities, are more similarly mapped in the models' hidden space, though the similarities are lower than for mutual translations (§3.1).

**Experiment 2: Patch-level analysis using logit lens.** We now test the image-description similarity using the logit lens, in Eq. 4. First, as a toy setting for illustration, we inspect LLaVA's representations of pure color images, specifically those in red, green, blue, and black. Figure 6b shows that, in up to more than 20% of the time in the intermediate layers (averaged across the patches and the four colors, $N = 2304$), the closest token is the corresponding color word (out of all vocabulary tokens).

We next consider the image captions, the same 1000. For each image patch, we compute a scalar patch-caption alignment score (for each layer separately) by summing over the logit lens probabilities for all nouns in its caption (as a proxy for objects in the image) at that image patch position. For the irrelevant token baseline, we compute the alignment score in the same way but using an unrelated caption. We normalize by the number of nouns so that the score is comparable. We average this alignment score over all patches in all images. Figures 7a and 7b show that the matching caption better aligns with the image patch representations than an unmatched caption, reliable across all layers for LLaVA and consistently in the middle-upper layers for Chameleon.

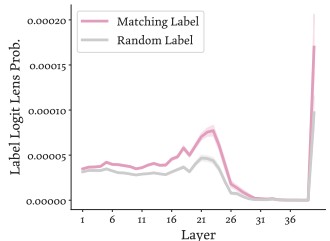

**Figure 8:** When SALMONN processes an audio clip, the logit lens prob. of the English words in the audio label vs. another random label. **The audio representations better match the semantically corresponding label in English.**

Finally, we perform a finer-grained study using not caption information but segmentation information, with object labels in specific image locations. The setup is near-identical captions, but the scalar alignment score for a given patch is not computed using its logit lens probability of caption nouns, but of its corresponding object label. For the irrelevant token baseline, we compute the alignment by aligning each patch with a different randomly chosen object category from all categories. Figures 7c and 7d show that, for LLaVA, the patches are much better aligned to the corresponding labels than randomly assigned labels (which have near-0 logit lens probability). For Chameleon, this is the case for only one middle layer, and not in a statistically significant way, though we will show in §4 that Chameleon's latent space can be reliably steered using English tokens.

## 3.4 AUDIO

Audio is another modality that is often modeled jointly with text (Lu et al., 2024; Gong et al., 2024; 2023; *i.a.*), and we perform similar experiments using SALMONN, an audio-text model. We use the VGGSound dataset (Chen et al., 2020) which contains 10-second audio clips with labels, e.g., "duck quacking" or "playing cello". We use the same two multimodal tests as in the vision case.

**Table 1:** Steering Llama-3's output sentiments using trigger words in English vs. the input language (either Spanish or Chinese). We report the mean sentiment, disfluency (perplexity), and relevance of the continuation, as well as the standard deviation across 10 seeds. **Cross-lingual steering is consistently successful, sometimes even more than monolingual steering, without substantial damage in text fluency and relevance.**

| Text Lang. | Steering Dir. | Steering Lang. | Sentiment | Disfluency ($\downarrow$) | Relevance ($\uparrow$) |
|---|---|---|---|---|---|
| Spanish | None | None | $0.143_{\pm 0.022}$ | $7.35_{\pm 1.19}$ | $0.861_{\pm 0.002}$ |
| | $\downarrow$ | Spanish | $0.125_{\pm 0.034}$ | $10.54_{\pm 2.39}$ | $0.842_{\pm 0.004}$ |
| | | English | $0.139_{\pm 0.026}$ | $8.75_{\pm 2.20}$ | $0.857_{\pm 0.002}$ |
| | $\uparrow$ | Spanish | $0.175_{\pm 0.035}$ | $7.98_{\pm 2.04}$ | $0.856_{\pm 0.002}$ |
| | | English | $0.159_{\pm 0.026}$ | $7.35_{\pm 1.01}$ | $0.859_{\pm 0.003}$ |
| Chinese | None | None | $0.178_{\pm 0.030}$ | $11.06_{\pm 3.12}$ | $0.869_{\pm 0.004}$ |
| | $\downarrow$ | Chinese | $0.152_{\pm 0.040}$ | $10.78_{\pm 2.66}$ | $0.866_{\pm 0.005}$ |
| | | English | $0.161_{\pm 0.029}$ | $11.36_{\pm 1.13}$ | $0.864_{\pm 0.004}$ |
| | $\uparrow$ | Chinese | $0.153_{\pm 0.034}$ | $11.12_{\pm 3.12}$ | $0.870_{\pm 0.004}$ |
| | | English | $0.179_{\pm 0.032}$ | $10.90_{\pm 3.25}$ | $0.869_{\pm 0.003}$ |

**Experiment 1: Representation similarity between audio and its label.** We study the representation cosine similarity between an audio and its label description, and subtract from it a baseline which is the average cosine similarity between non-matching pairs, separately for each layer. On 1000 samples from VGGSound, we see in Figure 6c that semantically matching audios and labels have more similar representations in the intermediate layers.

**Experiment 2: Token-level analysis using logit lens.** Unlike for vision-language models where we can map individual image patches to model token positions, such correspondence does not exist in SALMONN. This limits us to position-agnostic evaluations like the captioning study, preventing fine-grained analysis such as segmentation. Similar to the captioning experimental design, we measure the average logit lens probabilities of the words in the label, and consider a random label in the dataset with no word overlap as the baseline. On the same 1000 samples, Figure 8 shows a familiar trend, where the audio hidden states are closer to semantically corresponding label words. We note that this is a lower bound—many words in some labels, such as the prepositions in the label "`writing on blackboard with chalk`", are unlikely to be represented in the audio hidden states.

## 4 Intervening in the Semantic Hub

Interpretability results should be tested under a causal framework to ensure that the observation is not a vestigial byproduct of model training with no effect on model behavior (Vig et al., 2020; Ravichander et al., 2021; Elazar et al., 2021; Chan et al., 2022; *i.a.*). Here, we show that the semantic hub does causally affect model output: Semantically transforming $\tau^{z^*}$ in English interpretably leads to corresponding behavior changes in the non-dominant data type. §B reports relevant hyperparameters.

**Multilingual.** Past work has shown that (monolingual) interventions in the middle layers can steer LM outputs predictably (Subramani et al., 2022; Turner et al., 2024; Rimsky et al., 2024; *i.a.*). If English-dominant LMs have a shared representation space, we should be able to intervene on it in English even with non-English inputs. We use a common intervention method, Activation Addition (ActAdd; Turner et al., 2024), which has two stages: (1) taking a steering word (and optionally a contrastive one) that semantically represents the steering effect, passing it through the same model, and taking its hidden states at an intermediate layer; (2) adding the steering hidden states to those in the original forward pass of the regular generation process, at the same layer. We take their sentiment-steering experiment but generalize it cross-lingually. See Turner et al. (2024) for details.

We consider two non-dominant languages, Spanish and Chinese, and take 1000 texts in the InterTASS dataset (Spanish; Díaz-Galiano et al., 2018) and the multilingual Amazon reviews corpus (Chinese; Keung et al., 2020), and generate continuations either without modifications or intervened using ActAdd. As the steering vector, we use the difference between a positive sentiment trigger word and a negative one, in the appropriate direction for negative or positive steering. Specifically, we use "`Good`" and "`Bad`" for English, "`Bueno`" and "`Malo`" for Spanish, and "好" and "坏" for Chinese. In addition to sentiment evaluation, we also measure the generation fluency and compute the relevance

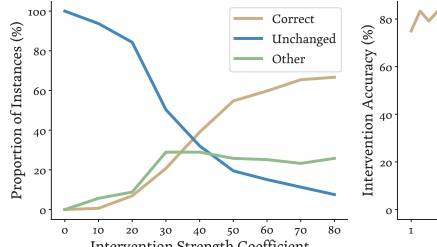 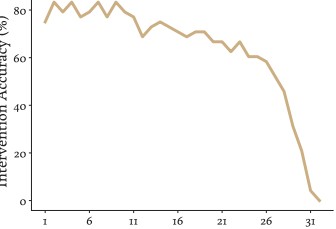 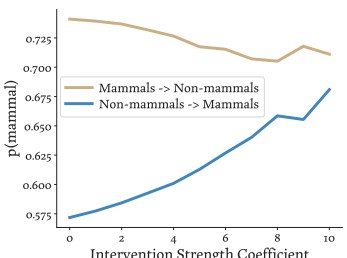

**(a)** Steering a single-argument "`range(end)`" call to be predicted as double argument.

**(b)** Replacing image representations of a color with language tokens of another color, and expecting the model to predict the new color.

**(c)** Steering mammal sounds to be predicted as non-mammal sounds, and vice versa.

**Figure 9:** For (a) code, (b) images, and (c) audio, steering model output using English words, for various intervention strengths ((a) and (c)) and layers ((b)). (a) and (b) measure successfulness with the proportion of instances steered to the correct output, and (c) measures the probability of predicting mammals. **Overall, intervening in the unified representation space in English reliably leads to desired model output changes.**

of the generation with the prefix using trained models, following Turner et al. (2024). Ideally, the intervention should achieve the desired sentiment without hurting text fluency/relevance (see §B.1).

Because we take intermediate layer representations of the steering words (step (1)), if the semantic hub is language-agnostic, we expect similar representations and similar steering effectiveness across steering languages. Table 1 shows that this is indeed true on Llama-3: ActAdd in the text language is in most cases effective, achieving the intended effect on sentiment, with usually only a modest decrease in fluency and relevance, often statistically insignificant. This aligns with the English-only findings in Turner et al. (2024). And intervening in English is similarly effective as using the text language. Table 2 (appendix) shows the results for Llama-2, with very similar trends.

**Code.** Based on our semantic role observation in §3.2, we run a intervention experiment using the "range" function. We focus on two overloaded versions of "range": "`range(start, end)`" and "`range(end)`". If the semantic hub causally affects model output, then intervening in it using the semantic role tokens should affect the function version used in a given context.

We take all single-argument "`range(end)`" calls in the MBPP dataset ($N = 159$) and attempt to expand it into "`range(0, end)`". As the intervention, we use an even simpler method than ActAdd: because the unembedding vectors of the semantic roles are close to the intermediate hidden states, we simply compute the difference between the unembeddings of two contrasting trigger tokens ("`start`" – "`end`"), scale it by a constant coefficient, and add it to the hidden representation corresponding to the open parenthesis "`(`" at an intermediate layer (layer 17). For all these "`range`" call in the dataset, we let Llama-2[4] generate without and with intervention. Figure 9a shows that, with increasing intervention strength, more instances are successfully steered to "`range(0, end)`", up to 67%.

**Visual inputs.** We show that we can steer the output of vision-language models by intervening on image patches using language tokens and analyze how this affects textual output. We focus on Chameleon which showed a weaker trend in §3.3. Focusing on the color setup, if the representation of a color is similar between visual and language inputs, we hypothesize that we can *replace* the image hidden states corresponding to one pure color image patch with the unembedding of the language token for another color, and mislead the model to "perceive" the new color when asked about it. Note that replacing the hidden state is a more invasive intervention than addition. But there is one confounder: the intervened word may lexically bias the model to generate the same word, without any reasoning that incorporates the new color. To control for this, we show two colors in one image and only intervene at the positions corresponding to one color: if the intervention unconditionally and lexically biases the generation to the new color, this effect would (incorrectly) affect both colors.[5]

---

[4] We do not consider Llama-3 in this case because its default behavior usually generates "`range(0, end)`" in the first place, and it is unclear how to steer from "`range(0, end)`" to "`range(end)`".

[5] We tested settings that require more sophisticated reasoning such as asking for a country flag with the two colors, or asking about spatial relationships of the colors. They seem to be beyond the capability of Chameleon-7B—even without interventions, the model cannot answer the questions correctly.

We consider all color pairs using the same colors as in §3.3: red, green, blue, and black, and picking one color in the pair and intervene to a new third color ($N = 48$). As the intervention, we start from a layer $\ell$ and replace all hidden states at and after $\ell$ to be the unembedding of the new color minus the old color. We ask the model what the two colors in the image are, and only consider the intervention successful if the model answers both the new color and the other unintervened color correctly. Figure 9b shows the success rate across all $\ell$: it gets as high as above 80%.[6] We highlight that, for both this experiment and the earlier ones in this section, the interventions are not even necessarily guaranteed to lead sensible outputs, let alone correct ones.

**Audio.** We perform a similar intervention with SALMONN, with the same desideratum that the QA process should require reasoning rather than outputting the intervened token as-is. We consider 1000 animal sounds in the VGGSound dataset, specifically only single-word animals, and ask "`Is this animal a mammal?`" We intervene both on mammal sounds with a random non-mammal word and vice versa, in case the intervention only biases the model in a certain direction. We perform the invention similarly to the code case, adding the unembedding difference between the new trigger word and the original animal name, scaled by a constant, at layer 13. We measure the probability of the "`Yes`" token and the "`No`" token and compute the normalized "`Yes`" probability. Figure 9c visualizes the two cases across intervention strengths. As the strength increases, the model is more likely to predict in the steered direction, again demonstrating cross-data-type intervention effectiveness.

## 5 RELATED WORK

**Representation alignment between separately trained models.** A long line of work has investigated the representations of separately trained mono-data-type models, and showed that they can be aligned through a transformation. In the multilingual case, it has been found that separately trained word embeddings for different languages can be aligned (Mikolov et al., 2013; Smith et al., 2017; Cao et al., 2020; *i.a.*). Similarly, prior work has shown that visual representations and text representations from different models can be mapped together (Merullo et al., 2022; Koh et al., 2023; Maniparambil et al., 2024; *i.a.*). Huh et al. (2024) argued that these are possible because the different data modalities are projections of the same underlying reality. Our work, in contrast, looks at a *single* model that processes multiple input data types and finds that the resulting representations align, without needing a transformation. Xia et al. (2023) considered an objective that explicitly trains the model to increase such alignment, while we analyze how it organically emerges through autoregressive training.

**Representation evolution throughout layers.** Past work has analyzed the representation evolution throughout transformer layers, inspecting how it affects reasoning (Yang et al., 2024), factuality (Chuang et al., 2024), knowledge (Jin et al., 2024), etc. From another angle, work on layer pruning and early exiting also speaks to representation dynamics across layers (Gromov et al., 2024; Sanyal et al., 2024; *i.a.*). Mechanistically, Elhage et al. (2021), Merullo et al. (2024), Todd et al. (2024), Hendel et al. (2023), *i.a.*, more precisely characterized the representation changes algorithmically.

**Inspecting model hidden states.** We adopted the logit lens for its simplicity which brings few confounders. However, alternatives exist, usually requiring some training (Belrose et al., 2023; Ghandeharioun et al., 2024; Templeton et al., 2024; *i.a.*). They allow for more expressive explanations, though at the risk of overfitting. Similar methods have been developed for other modalities, such as Toker et al. (2024). Testing our hypothesis using these methods would be valuable future work.

## 6 CONCLUSION

We proposed and investigated the semantic hub hypothesis, which posits that LMs represent semantically similar inputs from distinct modalities near one another in intermediate layers. We find evidence of this phenomenon across multiple LMs and data types, and further observe that intervening in this space in the model's dominant language (usually English) leads to predictable behavior changes.

---

[6]One may argue this is conceptually similar to a half-language half-image input. There are many distinctions: most importantly, a half-image is not processable by Chameleon and severely goes out of its training distribution, since it only ever processes images of size exactly $512 \times 512$. Other distinctions include: the presence of a special token marking the beginning of the image; our intervention repeats the new color token, once for each patch, rather than just one; and the token representation is held constant across layers rather than evolving; etc.

ACKNOWLEDGMENTS

We thank Alex Gu, Alexis Ross, Alisa Liu, Aryaman Arora, Asma Ghandeharioun, Cedegao E. Zhang, Freda Shi, Han Guo, Jack Merullo, Linlu Qiu, Mor Geva, Naman Jain, Ruochen Zhang, Sarah Wiegreffe, Shushan Arakelyan, and Yung-Sung Chuang for discussions and help at various stages of this project. Figure 1 uses icons from `flaticon.com`. This study was supported by funds from MIT-IBM Watson AI Lab.

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

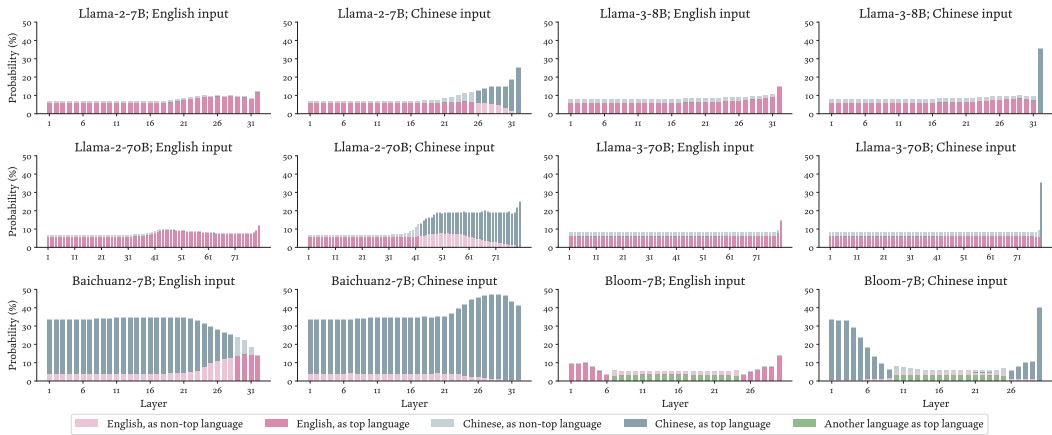

**Figure 10:** Language probabilities of English and Chinese (and the top language, when it is neither, which only happens for Bloom). **Regardless of the input language, the dominant LM language is more salient in the early-middle layers, and the input language is more salient in the final layers. Bloom does not have a clear intermediate latent language.**

## A  EXPERIMENTAL DETAILS FOR §3

### A.1  MULTILINGUAL

For Experiment 1, for each sentence pair, we use a template to transform each sentence. This is due to the automatic code-switching behavior of LMs. For an English model processing Chinese text, we expect the Chinese tokens to have high probabilities in the final layer because they need to be output; however, we observe these models tend to code-switch back to their dominant language after a full sentence, which confounds our analysis. We therefore put the parallel sentences into a template, "{English Sentence} This represents" (and the corresponding Chinese version), as the model is less likely to code-switch mid-sentence after "represents". We experimented with other templates that led to similar results. Furthermore, for sentence in GALE (Chen et al., 2016), we make sure the `transcript;unicode` is not empty for both the source and the translation.

For Experiment 2, due to tokenization, it is challenging to obtain exactly parallel English-Chinese tokens, and hence we perform aggressive filtering. We consider only text positions where the next BPE token (1) is a valid Chinese word (as segmented by Jieba (Sun, 2024)), and (2) has an English translation (using the English-Chinese dictionary CC-CEDICT (MDBG, 2024)). For example, "今天是开心的一天" (Today is a happy day), Llama-3 tokenizes it as {'今天', '是', '开', '心', '的一', '天'}, while Jieba segments it to be {'今天', '是', '开心', '的', '一天'}. We only keep {'今天', '是'}. Furthermore, only '今天''s translation is a single token, the only token that survives the cutoff is '今天'.

The full result of Figure 3 is in Figure 10. We observe that the 7B/8B and 70B models of the same model family have highly similar trends, so we only consider the 7B/8B models in other experiments.

### A.2  CODE

We consider all non-zero-argument function calls in the MBPP dataset, excluding unit tests. We automatically identify the argument names (the "semantic roles") by function inspection for built-in functions and by looking at the function definition for those defined in-context, and skip when this is not possible. We also ignore arguments whose semantic roles are generically called "`obj`" or "`object`", and instances where the instantiated surface-form argument is the same as the semantic role. We look the hidden state corresponding to the previous token, either "(" or ",", except when tokenization renders this impossible (e.g., when the previous token is merged with a part of the surface argument). This leaves 540 arguments.

**Table 2:** Steering Llama-2's output sentiments using trigger words in English vs. the input language (either Spanish or Chinese). We report the mean sentiment, disfluency (perplexity), and relevance of the continuation, as well as the standard deviation across 10 seeds. **Cross-lingual steering is consistently successful, sometimes even more than monolingual steering, without substantial damage in text fluency and relevance.**

| Text Lang. | Steering Dir. | Steering Lang. | Sentiment | Disfluency ($\downarrow$) | Relevance ($\uparrow$) |
|---|---|---|---|---|---|
| Spanish | None | None | $0.144_{\pm 0.014}$ | $8.58_{\pm 0.57}$ | $0.850_{\pm 0.006}$ |
| | $\downarrow$ | Spanish | $0.143_{\pm 0.012}$ | $8.84_{\pm 0.79}$ | $0.847_{\pm 0.006}$ |
| | | English | $0.097_{\pm 0.024}$ | $8.99_{\pm 0.72}$ | $0.847_{\pm 0.005}$ |
| | $\uparrow$ | Spanish | $0.164_{\pm 0.018}$ | $9.11_{\pm 0.50}$ | $0.844_{\pm 0.005}$ |
| | | English | $0.149_{\pm 0.015}$ | $8.35_{\pm 0.30}$ | $0.849_{\pm 0.006}$ |
| Chinese | None | None | $0.223_{\pm 0.036}$ | $14.63_{\pm 2.65}$ | $0.844_{\pm 0.009}$ |
| | $\downarrow$ | Chinese | $0.117_{\pm 0.080}$ | $15.29_{\pm 2.47}$ | $0.840_{\pm 0.011}$ |
| | | English | $0.156_{\pm 0.076}$ | $14.80_{\pm 2.24}$ | $0.842_{\pm 0.008}$ |
| | $\uparrow$ | Chinese | $0.359_{\pm 0.077}$ | $545.94_{\pm 1544.36}$ | $0.839_{\pm 0.010}$ |
| | | English | $0.227_{\pm 0.038}$ | $14.14_{\pm 2.42}$ | $0.845_{\pm 0.009}$ |

### A.3 VISION-LANGUAGE

To pass the images through the model, we embed them in templates, only for the logit lens experiments. For the color experiment, we use the template "USER: What is the color in the image?<image>\n ASSISTANT:". For the caption and segmentation experiments, we use "USER: What is in the image?\n<image> ASSISTANT:" for LLaVA and "What is in the image?\n<image>" for Chameleon.

For all caption and segmentation experiments, we use the MSCOCO 2017 dataset. For the caption evaluation, for each image patch (with the associated caption for the entire image), we compute a scalar patch-caption alignment score (for each layer separately) with nouns in the caption. We consider words with NOUN or PROPN tags given by SpaCy's en_core_web_trf model (Honnibal & Montani, 2017).

For the segmentation evaluation, we use the MSCOCO 2017 panoptic segmentation labels. The metric calculation is similar as above. Instead of a scalar patch-caption alignment score, we consider a scalar patch-label alignment score between a patch and its matching segmentation label, computed likewise using the logit lens. We consider a patch and a label as matching if there is an image segment with that label that occupies more than half of the pixels in the patch. Under this definition, a patch cannot have more than one label. When a patch is not matched with any label, we disregard it. In the baseline, we use a randomly chosen incorrect label from all possible labels for the alignment score. Finally, we average this alignment score across all patches in all images to obtain the curves in Figure 7c and 7d.

## B EXPERIMENTAL DETAILS FOR §4

For both the code and vision-language intervention experiments, we use argmax decoding.

### B.1 MULTILINGUAL

For each language, we sample $N = 1000$ instances from the training set of InterTASS for Spanish and the multilingual Amazon reviews corpus for Chinese. Following Turner et al. (2024), we use trained models for various metrics. We automatically evaluate the sentiment of the generation using a DistillBERT-based (Sanh et al., 2020) model finetuned for multilingual sentiment analysis,[7] judge the generation fluency by taking the conditional perplexity of the generation given the prefix from

---

[7] https://huggingface.co/lxyuan/distilbert-base-multilingual-cased-sentiments-student

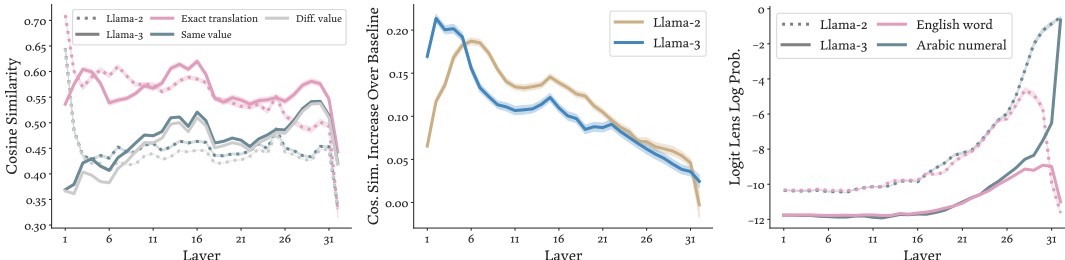

**(a)** Cosine similarity between an arithmetic expression in Arabic numerals vs. English words, broken down into separate categories.

**(b)** Same as (a), but only the exact translation similarities subtracted by the others.

**(c)** Logit lens log probability when predicting a number, between either the number itself or its English equivalent.

**Figure 11:** Results for the arithmetic experiments. The 95% CI is plotted in all. **Expressions in Arabic numerals have similar representation as corresponding expressions in English, as well as the unembeddings of corresponding number words in English.**

Llama-3.1-70B (Llama-3-Team, 2024),[8] and compute the relevance of the generation with the prefix by computing the cosine similarity between the generation and the prefix using a XLM-R-Large-based (Conneau et al., 2020) model finetuned for sentence representation (Wang et al., 2024). All these models support both Spanish and Chinese.

We perform ActAdd by passing both the positive and negative steering words through the LM, taking their hidden states at layer 17, computing their difference, scaling it by a constant, and adding it to the normal generation forward pass also at layer 17, exactly following Turner et al. (2024), except we use a scaling coefficient of 5, rather than 2 in their experiments, for which we observed a larger effect. For generation, we use a temperature of 1, top-$p$=0.3, and a frequency penalty of 1, all following Turner et al. (2024), without tuning.

We showed the Llama-3 intervention results in Table 1, and here in Table 2 we show the results on Llama2, with similar trends.

## C  EXPERIMENTS WITH ARITHMETIC EXPRESSIONS

We hypothesize that a similar trend exists when LMs process arithmetic expressions where they route to a shared space anchored by numerical words in English in intermediate layers. We consider simple expressions in the form of "a=b+c" or "a=b*c"; for simplicity, we restrict "a" and "b" to be at most two digits and "c" to be a single positive digit.

**Experiment 1: Representations are similar for translations.**  Here, we only consider the right-hand side, "b+c" and "b*c", as $w_{1:t}^{z_1}$ in Eq. 1. Like in the multilingual case, we translate them into English (e.g., "five plus three") as $w_{1:t'}^{z_2}$, and evaluate the representation cosine similarity between every English expression and every numeric expression, throughout layers. We group the pairwise cosine similarities in three buckets: (1) exact translation (e.g., "5+3" and "five plus three"; $N = 1123$), (2) non-exact but same value (e.g., "5+3" and "two plus six"; $N = 13293$), and (3) different value ($N = 1247836$). Figure 11a shows that exact translations have high cosine similarity, although this is to be expected since embeddings of numbers and their corresponding English words are near one another (thus even a bag-of-word-embeddings should also have high similarity). More interestingly, we that the similarities are still higher when the surface forms are distinct but the "meaning" of the expression (i.e., the value of the expression) is the same. Next, like in §3.1, we subtract the cosine similarities among non-translation pairs as a baseline ($u_{1:t'}^{z_2}$). Figure 11b shows high similarity in the early-middle layers for translations over the baseline, but gradually decreasing to near 0.

---

[8]We also tried using Mistral-Nemo-Base (https://huggingface.co/mistralai/Mistral-Nemo-Base-2407), and found similar trends.

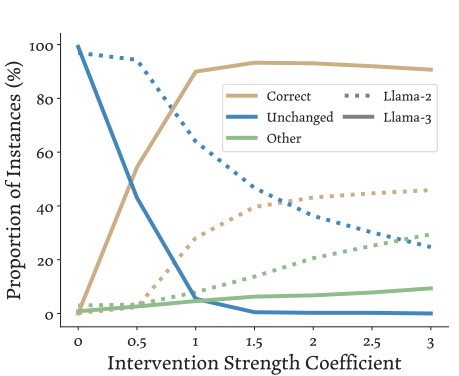
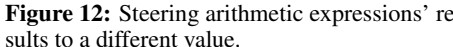

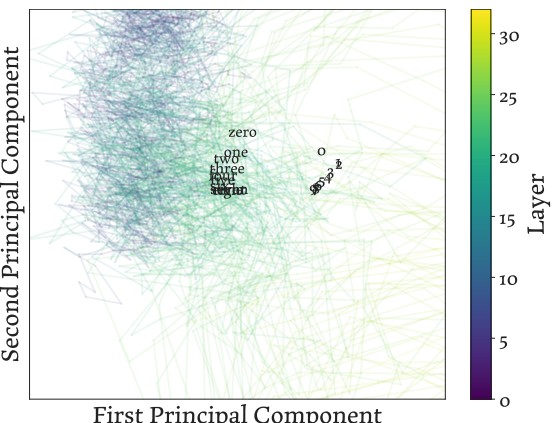

**Figure 12:** Steering arithmetic expressions' results to a different value.

**Figure 13:** The Llama-3 hidden representation evolution when predicting a number, projected by PCA where the principal components are learned on the output embeddings of 20 number tokens, 10 in English and 10 numerals.

**Experiment 2: Representations are anchored by semantically-equivalent English words.** We hypothesize that, for some prefix such as "a=b+", the intermediate representations $h_t^\ell$ are close to the English word for "c" that would make the equality hold. First, we randomly sample 100 such prefixes and take the representation of the last token at all layers. For each prefix, we plot the representation evolution throughout layers using PCA, as well as the unembeddings of numbers in English $\tau^{z^*}$ vs. numerals $\tau^{z^\circ}$. Figure 13 shows that the representations indeed go through the space occupied by the English words in intermediate layers. Next, we repeat our logit lens experiments, inspecting the log probability of the following numeral token vs. its English version ($N = 1123$). Figure 11c shows that the two tokens have similar log probability until around layer 25, after which the numeral token dominates.

**Experiment 3: Intervention.** We perform intervention using our arithmetic expressions, for example "4=1+3". We intervene by attempting to modify the token after "+" to be one smaller, e.g. "2" here, and expect this to not only lead the model to output "2" instead of "3", but also fundamentally affects the model's reasoning process and causes the model to patch this error with an additional suffix "+1", i.e., "4=1+2+1". We use ActAdd except for adding the intervention vector (e.g., "three" – "two") only at the position of "+".[9] For all addition expressions in our data ($N = 846$), we perform such intervention at an intermediate layer (25 for Llama-3 and 30 for Llama-2) and measure how often this leads to the model correctly outputting the decremented number followed by "+1", versus unchanged, or changed to some other output. Figure 12 shows that, as the intervention coefficient (i.e., the scaling constant of the vector) increases, this procedure leads to the expected output for up to $> 90\%$ of the instances.

---

[9]Another difference is that we do not use the hidden representation after seeing e.g. "three", because that usually represents the *next* token. Instead, we use a prefix that uniquely determines the number, e.g. "Eight equals to five plus", and take the last token hidden representation, which *is* supposed to represent "three".

