# OpenReview forum: "The Semantic Hub Hypothesis: Language Models Share Semantic Representations Across Languages and Modalities"
_ICLR.cc/2025/Conference — ICLR 2025 Poster_

### Official Review · Reviewer_MWcS · 2024-10-28

**Soundness:** 3
**Presentation:** 3
**Contribution:** 3
**Rating:** 6
**Confidence:** 3

**Summary:**

This paper investigates whether multimodal LLM  process diverse input types using a unified representation space. The authors utilized "logitlens" as an interpretability tool to extract representations from each layer of the LLM. They show that model representations for semantically equivalent inputs from different modalities are similar in intermediate layers and that interventions in this space lead to predictable changes in model behavior.

**Strengths:**

1.  Extends prior research (Wendler et al.) from multilingual to multimodal settings on unified representation in the middle layers of LLM, offering new insights.
2. The experiments are interesting and thorough across modalities, including text, code, image, audio, adding robustness to its claims.
3. Good experiments on intervention in the unified representation space which shows the space causally steers model output.

**Weaknesses:**

1. The paper is largely an empirical study, and the findings may heavily depend on experimental settings. Some details are unclear, e.g., in multilingual experiment 1, the baseline for non-parallel texts is not well explained.
2. The paper consistently use the last token's representations over layers for measuring similarities interpretation, which maybe a limitation.
3. In Figure 9, the cosine similarity increase over baseline is marginal, with a maximum of only 0.03.
4. In line 435, the paper hypothesizes that the unified representation space is language-agnostic, but the experiments primarily use the English-dominant Llama-3. This needs more justification.
5. Lack of related work for representation alignment, such as alignment through relative space [1] and its application in LLMs [2].

[1] Moschella et al. "Relative representations enable zero-shot latent space communication"
[2] Wu et al. "Zero-Shot Continuous Prompt Transfer: Generalizing Task Semantics Across Language Models"

**Questions:**

1. Why Bloom is not a english-dominant LLM but Llama is english-dominant? Does it mean that they were trained differently?

---

> ### Author Response · Authors · 2024-11-19
>
> Thank you for your thoughtful review and your appreciation of the interest value and the robustness of our experiments!
>
> **Weakness 1 (the dependence of results on experimental settings)**: While results of all experimental studies inevitably depend to some extent on the experimental settings, we use standard methods (e.g., the logit lens) and design decisions in our settings. In particular, we adopt the logit lens, a training-free method, for analysis precisely because it does not require learning a probe and minimizes potential confounders. We are confident that alternative experimental design decisions would lead to similar findings. Regarding the specific question about the non-parallel baseline: we randomly choose one non-matching sentence as the baseline. In the revision, we added this explanation, as well as additional experimental details in section A of the appendix. Thank you for this suggestion!
>
> **Weakness 2 (using the last token’s representation as the sequence representation)**: We have already included a discussion about this in L193-194, both citing prior work with the same practice, as well as more principled evidence that the last token representation faithfully retains sequence information. We also performed preliminary studies using average pooling of token representations along the sequence dimension, and found similar trends as taking the last token’s representation.
>
> **Weakness 3 (the small absolute scale of cosine similarities)**: This is indeed true, but we emphasize that absolute similarity measures are difficult to interpret in high dimensional spaces (where all vectors are nearly orthogonal to one another). Moreover, the confidence intervals indicate that the effect is statistically significant.
>
> **Weakness 4 (the use of English-dominant models)**: We primarily tested on English-dominant models because most state-of-the-art models are predominantly trained using English data. But we do test LLMs with other dominant languages, as well as those without a dominant language, in section 4.1.
>
> **Weakness 5 (related work)**: Thank you for bringing these papers to our attention! We have added citations and a brief discussion around them in our introduction (L89-91) in the revision.
>
> **Question 1 (the multilinguality of BLOOM)**: Yes, BLOOM was trained on a relatively balanced mixture of multilingual data and hence not English-dominant (see its Figure 3). On the other hand, Llama-2 and Llama-3 were trained on predominantly English data.

---

### Official Review · Reviewer_nLRX · 2024-11-02

**Soundness:** 4
**Presentation:** 4
**Contribution:** 3
**Rating:** 8
**Confidence:** 4

**Summary:**

This paper proposes the Unified Representation Hypothesis, which predicts that—regardless of the type of data of the inputs being processed—the latent representations of a given concept will be aligned with the dominant data type that the model was trained on. For example, given a multimodal language model trained primarily on English text, the latent representations will reliably align with English text, even if the input is only an image, or in another language.

The study tests and finds consistent support for this hypothesis in cross-lingual and cross-modal settings, including text-audio, text-image, and multilingual models. These alignments are largely verified using the logit lens, but many experiments also entail counterfactual interventions to hidden states.

**Strengths:**

* The study advances an idea that has been more casually discussed in past work: that the representation space of models representing many types of data will be biased toward the dominant data type. This paper proposes and presents evidence for a stronger version: that, when given data of any type, the model’s representation of what to predict next will be more closely aligned with the dominant data type (even in the absence of any data of the dominant type in the input).
* The ideas are presented in a precise way. While it may take more effort to parse than verbal intuitions, it makes the theoretical assumptions underlying the work clearer.
* Thorough experimentation. Convincing evidence (often causal) from many models and in many settings.
* Table 2: It’s nice that the quality of the outputs is evaluated along multiple axes, rather than just overlap with respect to a reference.

**Weaknesses:**

* The proposed idea does not seem distinct from that proposed in the Platonic Representation Hypothesis paper (Huh et al., 2024), which is cited. That paper also presented similarly diverse and thorough empirical evidence in favor of unified representations across modalities. It would be helpful to explicitly contrast the Unified Representation Space Hypothesis from the Platonic Representation Hypothesis in the paper, or at least to contextualize the main idea and findings here more thoroughly with it.
* Due to the quantity of experiments in the paper, it was probably difficult to fit the necessary details for each in such a way that makes them all clear. Most of them are well-defined, but I was sometimes not sure what, exactly, was being evaluated. See Questions/Suggestions for details.
* L108-L113: Relative similarity captures a notion of distance, but it doesn’t capture whether there is underlying structural similarity (e.g., isotropy). Token distance is a nice start, but I have a hunch that going deep rather than wide would yield even more significant insights.
* L257-258: Speculative. It would be interesting to test this by projecting the inputs onto the space of the dominant data type, and then generating.

**Questions:**

* Figure 11: The experimental setup should be clarified. Is this a single comparison of the similarity between the logit lens and the mean representation across all nouns in the caption? Or is this an average similarity across separate similarities per noun? Or something else?
* L360-368: This is pretty light on details.  appendix explaining the experimental settings would be very helpful.
* Figure 8: There is plenty of vertical space here; consider enlarging this figure. Also: this looks pretty noisy. It could be revealing to add a color for a control setting, where the hidden representation is closer to a token other than these two possibilities. I hypothesize that much of this graph will be filled with the control color, but also that the rows with blue color will be much more consistent across examples.
* Figures 12 and 13: These values seem small on an absolute scale, and the difference between baseline and the correct answer is also small on a relative scale.

---

> ### Author Response · Authors · 2024-11-19
>
> We are grateful for your review and are glad to see that you appreciate the technical framework underlying our experiments, as well as the thoroughness of the experiments.
>
> **Weakness 1 (comparison with the Platonic Representation Hypothesis paper)**: Huh et al. (2024) is certainly related to our work, and we discussed this relationship in the final paragraph of section 1 and the first paragraph of section 4.3. In particular, their claim is that **_different_** models of the same task, or even of different tasks (more relevant to us), have converging representations. In contrast, we consider the representation similarity within a **_single_** multimodal model. Our notion of representation convergence is stronger than theirs: in their experiments, they found that the model representations are isomorphic (e.g., the distance between the words “dog” and “cat” are similar to the distance between pictures of a dog and a cat), while we directly look at cross-modal representation similarity (e.g., the representation of the word “dog” is close to that of the picture of a dog). We clarified this a bit more in the final paragraph of section 1 in the revision. Other than this explicit operationalization, Huh et al. (2024) also discusses such convergence on a more philosophical level. We added a reference to this discussion in the related work section.
>
> **Weakness 2 (paper organization)**: Thank you for this feedback! We will certainly improve the paper organization in the next revision, putting some content into the appendix as needed. We respond to your specific suggestions below.
>
> **Weakness 3 (limitation of our similarity measure)**: We agree that structural similarity would be a stronger measure than pointwise relative similarity, but we believe they are correlated. In the extreme, if Equation 1 were satisfied for all possible inputs, that would imply complete structural similarity. Measuring structural similarity may require design decisions that potentially confound the results (for example, Huh et al. (2024) entertained multiple kernel-alignment metrics). But in any case, we do think a more in-depth investigation of our observations would be exciting future work. We prioritized investigating a variety of data types because we found our observations very interesting and wanted to reliably verify that this is a consistent and robust property of models across data types, before more closely analyzing this property in future work.
>
> **Weakness 4 (suggestion of projecting inputs to the dominant data type and then generate)**: We argue that this is not a weakness per se, but it is certainly an interesting suggestion! We set up our experiments to enable an investigation of the hypothesis in a training-free setup which would not require learning an alignment model to go between modalities, in order to minimize potential confounds. But a more heavily parameterized probe model could uncover even more interesting phenomena and it would certainly be an interesting experiment to run as future work!
>
> **Question 1 & 2 (multimodal alignment score calculation details)**: Thanks for pointing out this potential source of unclarity. We added more detailed descriptions of the metrics in appendix A.3.
>
> **Question 3 (figure for the code semantic role experiment)**: We implemented your suggestion about control tokens. In https://ibb.co/SxbdZ5h, we show the same plot, except that we make the cell gray when the probability of neither token (the semantic role and the surface argument) is greater than the whitespace as the control token. The noise still persists since there are not many gray cells; this is still roughly true when we use the word “object” (https://ibb.co/dM6BGGC) or “.” (https://ibb.co/K508BC9) as the control token. Stepping back, instance-level variance (that leads to the noise) is in some sense inevitable, but we believe the blue/red separation is nonetheless clear from our plot. Also, thank you for the presentation suggestion! We improved this slightly in the updated version and will systematically improve the presentation of all plots in our next revision.
>
> **Question 4 (the small absolute scale of cosine similarities)**: This is indeed true, but we emphasize that absolute similarity measures are difficult to interpret in high dimensional spaces (where all vectors are nearly orthogonal to one another). Moreover, the confidence intervals indicate that the effect is statistically significant.

---

> > ### Comment · Reviewer_nLRX · 2024-11-20
> > **Response**
> >
> > Thank you for the detailed response! Weakness 1 and Questions 1 and 2 have been addressed. It sounds like Weakness 2 will be addressed in the final version, so I will hold off on considering this addressed. In particular, it would be nice to find a way to move the response to Questions 1 and 2 to the main paper, and move something else to the appendix (perhaps one of the follow-up analysis experiments); these are important details!
> >
> > Thanks for running the experiment for the Question 3 response. It's unfortunate that this still looks noisy, but the trends across instances are still revealing anyway.
> >
> > As my current score is quite positive, I'm choosing to keep this the same for now. I would be happy to reconsider the overall score and increase the presentation score given a revised PDF that makes the requested organization and presentation changes.

---

> > > ### Author Response · Authors · 2024-11-22
> > >
> > > Thank you very much for your reply! As you suggested, we have moved the majority of these experimental details into the main text (L350-365), only keeping minor details in the appendix such as how we determine the noun-ness of a word. We have also updated the presentation of all of our figures, improving, among many other things, the vertical space surrounding Figure 8 (now Figure 5(b)).

---

> ### Comment · Reviewer_nLRX · 2024-11-24
> **Response**
>
> Thanks! Weakness 2 can now be considered addressed as well. Upping my presentation score. I believe the authors have done a good job of addressing the weaknesses I have raised as well as those raised by other reviewers. There are some small outstanding weaknesses from me and others; however, I would really like to see this paper at ICLR and future work in this vein! I wish there were an option to give a 9, and would give this score if it were an option. Please consider my review on the more positive side of an 8.

---

> > ### Author Response · Authors · 2024-11-25
> >
> > Thank you for the updated score! We are very grateful for your time and the valuable discussion.

---

### Official Review · Reviewer_EcEH · 2024-11-03

**Soundness:** 3
**Presentation:** 3
**Contribution:** 2
**Rating:** 6
**Confidence:** 4

**Summary:**

The paper proposes a hypothesis that models can learn a unified representation space across multiple modalities, such as language and vision, as well as across multiple languages. Specifically, the authors hypothesize that: (1) Semantically related concept pairs from different modalities or linguistic spaces are represented more closely than unrelated pairs. (2) When given an input, the model continues in the dominant modality (e.g., language for a language model), even when the input format is symbolic, such as code.

To validate the first hypothesis, the authors calculate the cosine similarity between paired data (e.g., parallel translations, images and captions) and non-paired data, observing that semantically paired data have higher similarity scores. For the second hypothesis, they evaluate whether tokens that match the input context have a higher probability than random tokens. The results suggest that the model favors continuations in the dominant modality.

The authors further examine cross-modal correspondence by intervening in one modality and observing changes in others, covering multilingual, code, image, and video data. For example, by replacing image hidden states corresponding to one color with the text embedding for another color, they find that the model correctly predicts the new color with 80% accuracy when replacements are made from the first layer onward.

In conclusion, the authors present empirical evidence through similarity analysis and cross-modal interventions to support that models can learn a semantically unified space across languages and modalities.

**Strengths:**

1. The topic of interpretability in multi-modal representations is highly relevant, and the hypotheses and conclusions presented in this paper—along with the supporting experiments—are likely to inspire further research.
2. The paper is well-written and easy to follow.
3. The authors validate their hypotheses across multiple modalities, including multilingual settings, text-code, language-vision, and language-audio, providing a thorough examination of their proposed unified representation space.

**Weaknesses:**

1: Sub-hypothesis 1, “semantically related concepts are closer than unrelated concepts,” is not novel, as it aligns with the standard training objective of many multi-modal models. This alignment is already an expected outcome, though recent research has highlighted its limitations and proposed methods to improve it [1, 2, 3].

2: Sub-hypothesis 2 suggests that inputs will continue in the dominant space, even if the input format is not in that space. The authors support this by observing that, in models like Llama, intermediate representations shift to the dominant language (e.g., English) when the input is Chinese, with the representation returning to Chinese dominance in later layers. However, this hypothesis seems unrelated to the main idea of a unified representation space, and its findings are largely anticipated. A deeper analysis is needed, potentially identifying specific transformation mechanisms, such as attention heads or MLPs, that facilitate cross-lingual translation. Insights into how these transformations occur would add value, as explored in the Induction Head blog[3], which discusses how output circuits transform intermediate representations into the output space.

3: Lack of discussion of related work in unified representation[1,2] and information flow, especially the intermediate representation to input/output space [3]

References:

[1] Achieving Cross Modal Generalization with Multimodal Unified Representation

[2] UNIFIED-IO: A Unified Model for Vision, Language, and Multi-modal Tasks

[3] A Mathematical Framework for Transformer Circuits

**Questions:**

1. The description of Figure1, and Figure2 is missing?

2. In Figure 6, the intermediate tokens shown are predominantly English words rather than the matched label token. Could the authors clarify whether the tokens from the last layer align with the matched token? If so, the results may not be surprising, as the intermediate layers likely serve to associate the input with internal knowledge in the dominant space, while the later layers have output circuits that transform this dominant representation back into the output format. This aligns with my earlier point about the expected behavior of the model’s translation mechanism (see Weakness Point 2)?

---

> ### Author Response · Authors · 2024-11-19
>
> Thank you so much for your detailed review! We are really happy that you recognize the timeliness and the potential future impact of our work, as well as the thoroughness of our experiments.
>
> **Weakness 1 (the expectedness of our results)**: While there are indeed multimodal training objectives that encourages the alignment of semantically equivalent cross-modal inputs, most commonly in a contrastive manner (e.g., CLIP [4]), these models are not a subject of our study. The multimodal models that we analyzed, LlaVA, Chameleon, and SALMONN, were all trained on a purely next-token prediction objective, *without* explicit token-level alignment of modalities (and yet we find that this alignment implicitly emerges). In particular, Chameleon was trained on mixed-modal data from the onset of pretraining, allowing the model complete flexibility in its internal representation, and hence we believe the cross-modal representation similarity that we observe is remarkable and not *a priori* expected. Likewise, in the multilingual and code experiments, there are no explicit training objectives for such alignment either. In the revision, in section C in the appendix, we also added a new set of experiments on an additional data type, arithmetic expressions (e.g., “5=3+2”). We found the same trend where LMs implicitly represents the numbers in English (e.g., “two”), even when there is again no explicit pretraining objective for such alignment. The intervention experiment also works analogously as in the other cases.
>
> **Weakness 2 (the relatedness of our two hypotheses)**: We disagree that our sub-hypothesis 2 is unrelated to the unified representation space. The fact that, for example, both English and Chinese inputs have intermediate representations close to the same English continuation token (sub-hypothesis 2) implies that the intermediate layers are indeed representing semantically similar inputs near one another. We moreover disagree that the findings are largely anticipated---Wendler et al. [5] showed a similar phenomenon (only recently) for a highly synthetic setting---we demonstrate that this persists across more naturalistic text and other types of inputs such as code, images, and audio. Nevertheless, we do agree that a deeper analysis would be insightful. We prioritized confirming our findings on a variety of data types in this work to ensure it is a robust property of models. A more in-depth study using more sophisticated methods would certainly be interesting future work!
>
> **Weakness 3 (the discussion of related work)**: Thank you for suggesting these papers! We did cite Unified-IO [2] as well as its follow-up Unified-IO 2 [6] (L319). We did not compare to [1] because they explicitly trained models to encourage such representation similarity while we analyzed how this similarity organically emerges implicitly through autoregressive pretraining. This is related to our response to Weakness 1 above. But we agree that this is an important point worth clarifying and we added a citation to [1] and a discussion in the Related Work section (section 6; L515-516) in the revision. Regarding “information flow, especially the intermediate representation to input/output space”, we do have a paragraph in our related work section dedicated to “Representation evolution throughout layers”, in particular with many papers in the same line of work as [3]. We have added a citation to [3] too in our revision.
>
> **Question 1 (figure caption display issue)**: We do not see this issue on our end. This could be a PDF rendering issue; using a different PDF viewer may help with this. We have also changed the figure format (PDF -> PNG) hoping to fix this. Could you help check if the issue still exists?
>
> **Question 2 (clarification regarding the code examples figure)**: In most cases, the final layer prediction indeed matches the groundtruth next token (as a good code model would), though not always. Regarding the surprisingness of our results, we emphasize that while having this common cross-modal knowledge is an intuitive representation strategy, there is no *a priori* reason that models adopt it. We believe it is valuable to investigate this phenomena across a wider set of settings than previously investigated. Furthermore, past work has only “casually discussed” this idea, as written by Reviewer nLRX. It is valuable that we concretely and mathematically defined and tested this hypothesis, some of which is arguably stronger than an abstract notion of shared knowledge, especially the causal intervention experiments.

---

> > ### Author Response · Authors · 2024-11-19
> >
> > [1] Yan Xia, Hai Huang, Jieming Zhu, and Zhou Zhao. Achieving cross modal generalization with multimodal unified representation. In Thirty-seventh Conference on Neural Information Processing Systems, 2023.
> >
> > [2] Jiasen Lu, Christopher Clark, Rowan Zellers, Roozbeh Mottaghi, and Aniruddha Kembhavi. UNIFIED-IO: A unified model for vision, language, and multi-modal tasks. In Proceedings of the International Conference on Learning Representations (ICLR), 2023.
> >
> > [3] Nelson Elhage, Neel Nanda, Catherine Olsson, Tom Henighan, Nicholas Joseph, Ben Mann, Amanda Askell, Yuntao Bai, Anna Chen, Tom Conerly, Nova DasSarma, Dawn Drain, Deep Ganguli, Zac Hatfield-Dodds, Danny Hernandez, Andy Jones, Jackson Kernion, Liane Lovitt, Kamal Ndousse, Dario Amodei, Tom Brown, Jack Clark, Jared Kaplan, Sam McCandlish, and Chris Olah. A mathematical framework for transformer circuits. Transformer Circuits Thread, 2021.
> >
> > [4] Alec Radford, Jong Wook Kim, Chris Hallacy, Aditya Ramesh, Gabriel Goh, Sandhini Agarwal, Girish Sastry, Amanda Askell, Pamela Mishkin, Jack Clark, Gretchen Krueger, Ilya Sutskever. Learning Transferable Visual Models From Natural Language Supervision. In International Conference on Machine Learning (ICML), 2021.
> >
> > [5] Chris Wendler, Veniamin Veselovsky, Giovanni Monea, and Robert West. Do llamas work in English? on the latent language of multilingual transformers. In Proceedings of the Annual Meeting of the Association for Computational Linguistics (ACL), 2024.
> >
> > [6] Jiasen Lu, Christopher Clark, Sangho Lee, Zichen Zhang, Savya Khosla, Ryan Marten, Derek Hoiem, and Aniruddha Kembhavi. Unified-io 2: Scaling autoregressive multimodal models with vision language audio and action. In Proceedings of the IEEE/CVF Conference on Computer Vision and Pattern Recognition (CVPR), 2024.

---

> ### Comment · Reviewer_EcEH · 2024-11-28
>
> Thank you for your response, especially for adding the figure description in the main content (rather than the caption) and enhancing the discussion on related work. Based on these improvements, I have raised my soundness score to 3 and my overall rating to 6.

---

> > ### Author Response · Authors · 2024-11-28
> >
> > Thank you so much!

---

### Official Review · Reviewer_id3E · 2024-11-04

**Soundness:** 3
**Presentation:** 4
**Contribution:** 4
**Rating:** 8
**Confidence:** 4

**Summary:**

This paper aims to show that models process inputs from different languages/modalities primarily by first projecting them into a unified representation space, before projecting them out into their language/modality for the output. They do so for multilingual, code, visual and audio settings. They then perform interventions in the unified space across those settings to support the hypothesis that the models operate in that space in intermediate layers.

**Strengths:**

* The paper builds on Wendler et al. (2024), which only looked at Llama trained pre-dominantly on English, to show that models trained predominantly on other languages do indeed have that language as their intermediate representation space
* They show that this also extends to multimodal input, which is what supports the claim of a 'unified' representation space.
* They perform some intervention experiments to support the hypothesis, and show that intervening in the unified representation space can affect outputs in other modalities/languages.

**Weaknesses:**

* In Line 108, they mention that 'absolute similarity measures are generally difficult and unintuitive to interpret in high dimensional spaces' and hence choose relative similarity measures instead. This is a key choice in the experimental setup, and I would hope to see more discussions on whether prior work have chosen to use relative over absolute similarity measures.
* The intervention experiments seem to be somewhat weak. For them to strongly support the unified representation space hypothesis, I would expect a comparison showing that  dominant data type steering is comparable or more effective than non-dominant steering. This doesn't seem to be the case as monolingual steering seems to be on the whole better than crosslingual steering (Table 1). However, I do think that the fact that English steering works at all weakly suggests that hypothesis is true.

**Questions:**

* Section 2: Notation wise, it is not clear what $M_{LM}$ is. It strictly reads as just the embedded tokens in this section (L102), but it also looks like a general, vague reference to mid-layer representations later in the paper.
* Equation 1 needs to be explained: is this the formulation of your hypothesis? If so, it should be introduced like: "Formally, our hypothesis can be formulated as:"
* What do you mean by "scaffolded" as used throughout the paper? The term is not defined but used early on in the introduction.
* Typos: (1) L288: Should this refer to Figure 8? (2) All references to Llava should be 'LLaVA' as per the original paper.

---

> ### Author Response · Authors · 2024-11-19
>
> Thank you for your thoughtful review! We are glad that you recognize the value of our paper.
>
> **Weakness 1 (the use of relative similarity measures)**: In our revision, we added a citation to a theoretical reason for this practice, as well as a discussion on its implicit use in prior work (footnote 2, L161). Specifically, similarity measures in high dimensional spaces are difficult to interpret on their own (where most vectors are orthogonal to one another) without a baseline with which to compare against. Most prior work on probing (including logit-lens probes) implicitly uses relative similarity measures since the similarity scores are normalized over a finite label set. However, this is difficult to do in our case because the set of "negative examples" is too large to normalize over. Additionally, we do show absolute similarity measures for some of our studies, for example in the mutual translation setting, which are still high.
>
> **Weakness 2 (multilingual intervention results)**: We agree that a strong version of the hypothesis would entail that dominant-language (i.e., cross-lingual) steering should outperform the monolingual setting. However, we believe that cross-lingual steering is at least comparable to the mono-lingual setting in Table 1. For example, to positively steer in Chinese, using English steering words outperforms using Chinese steering words. For the other cases, while monolingual steering leads to larger effects, it also comes with larger degradation of fluency and relevance. This is not necessarily “better” because this could also be achieved by simply increasing the intervention strength coefficient (which we set to the value in Turner et al. (2024) without case-by-case tuning). Stepping back, we agree with you that the fact that English steering works at all is evidence for the hypothesis.
>
> **Questions (minor presentation issues)**: Thank you for raising these issues! We have clarified/fixed all of them in the revision. Specifically: (1) $M_{LM}$ is meant to abstractly denote an intermediate representation generated by early LM layers (not necessarily just the embedding layer; clarified in L105-106); (2) Equation 1 is indeed a formulation of our hypothesis (clarified in L114-115); (3) from our revision: “By scaffolded, we mean that the shared space can be interpreted to an extent in the dominant data type via the logit lens.” (L42-43); (4) L288 (now L291) actually refers to the second example in Figure 1; we have clarified this.

---

> > ### Comment · Reviewer_id3E · 2024-11-26
> >
> > I thank the authors for their response.
> >
> > Just a minor note: I think by using the terms 'scaffold' and 'dominant data type' in the introduction, the hypothesis now reads awkwardly in the current draft:
> >
> > > We show that this is in fact a much more general phenomenon:
> > > **when a model processes inputs from multiple data types, there is a shared representation space,
> > and this space is scaffolded by the LM’s inherently dominant data type (usually English).** By
> > scaffolded, we mean that the shared space can be interpreted to an extent in the dominant data type
> > via the logit lens (nostalgebraist, 2020).
> > """
> >
> > If you replace the word 'scaffold' with its definition in the first sentence, it reads as:
> > > "when a model processes inputs from multiple data types, there is a shared representation space, and this shared space can be interpreted to an extent in the dominant data type."
> >
> > Is this an accurate depiction of what you are trying to claim? I feel like what you have in the abstract, which is
> > > "We first show that model representations for semantically equivalent inputs in different languages are similar in the intermediate layers, and that this space can further be interpreted using the model’s dominant pretraining language (when it has one) via the logit lens."
> >
> > is a much better and straightforward way of putting across the same idea, without having to use phrases like "dominant data type" which can be ambiguous especially early on in the introduction.

---

> > > ### Author Response · Authors · 2024-11-26
> > >
> > > Thank you for this further suggestion! We have revised the main hypothesis statement and removed the use of "scaffolded" and "dominant data type" as you suggested.

---

### Author Response · Authors · 2024-11-22

We thank all reviewers for their detailed reviews! We are happy to see that the reviewers recognize the potential future impact of our findings and the thoroughness of our experiments that support the findings. We have uploaded a new revision that addresses the reviews. The changes are marked in green. The biggest updates are:
1. An additional set of experiments on a new data type, arithmetic expressions, in the appendix (section C). We found the same trend where, when LMs process arithmetic expressions in Arabic numerals (e.g., "5=3+"), they implicitly represent the numbers in English (e.g., “two”). The intervention experiment also works analogously as in the other cases.
2. We have updated the presentation of all of our figures.
3. We have clarified our experimental setup and added suggested references that reviewers suggested.

---

### Author Response · Authors · 2024-11-25

We really appreciate all the reviewers for helping review this paper. Given that the discussion period is ending tomorrow, we were wondering if the reviewers could let us know if our responses have addressed your concerns and questions about the paper? We would be happy to engage in further discussions if needed!

---

### Meta-Review · Area_Chair_CDFY · 2024-12-21

**Metareview:**

The paper introduces unified representation hypothesis, proposing that LLMs implicitly learn a unified representation space across multiple modalities and languages. The study presents empirical evidence through similarity analysis and cross-modal interventions, demonstrating how intermediate representations converge within this unified space. Reviewers commended the paper's thorough experimentation, covering diverse modalities such as text, code, image, and audio, and appreciated its methodological clarity and interoperability. I recommend accepting this paper.

**Additional Comments On Reviewer Discussion:**

During the rebuttal, the authors provided clarifications, conducted additional experiments to strengthen their claims, and revised the paper for improved contextualization and clarity. These efforts effectively addressed the reviewers' main concerns.

---

### Decision · Program_Chairs · 2025-01-22

Accept (Poster)